# Smart Manufacturing Scheduling Approaches—Systematic Review and Future Directions

**Duarte Alemão** [1,2,*] , **André Dionisio Rocha** [1,2] and **José Barata** [1,2]

1 UNINOVA—Centre of Technology and Systems, FCT Campus, Monte de Caparica,
2829-516 Caparica, Portugal; andre.rocha@uninova.pt (A.D.R.); jab@uninova.pt (J.B.)
2 Department of Electrical and Computer Engineering, Faculty of Sciences and Technology, NOVA University
of Lisbon, 1099-085 Lisbon, Portugal
* Correspondence: d.alemao@uninova.pt

**Abstract:** The recent advances in technology and the demand for highly customized products have been forcing manufacturing companies to adapt and develop new solutions in order to become more dynamic and flexible to face the changing markets. Manufacturing scheduling plays a core role in this adaptation since it is crucial to ensure that all operations and processes are running on time in the factory. However, to develop robust scheduling solutions it is necessary to consider different requirements from the shopfloor, but it is not clear which constraints should be analyzed and most research studies end up considering very few of them. In this review article, several papers published in recent years were analyzed to understand how many and which requirements they consider when developing scheduling solutions for manufacturing systems. It is possible to understand that the majority of them are not able to be adapted to real systems since some core constraints are not even considered. Consequently, it is important to consider how manufacturing scheduling solutions can be structured to be adapted effortlessly for different manufacturing scenarios.

**Keywords:** manufacturing scheduling; smart manufacturing; intelligent manufacturing systems; scheduling requirements; cyber-physical production systems



## 1. Introduction

Manufacturing has suffered deep changes over the past decades, mainly driven by the market trends that forced companies to move from traditional mass production lines to more dynamic and flexible manufacturing systems. The increasing demand for highly customized products with several variants led to smaller lot sizes, which requires companies to quickly adapt and adjust to new market opportunities in order to thrive in a very competitive world. Therefore, it is crucial that manufacturers develop approaches that allow for more dynamism, flexibility, and reconfigurability at the factory level.

The life cycles of product are becoming smaller and smaller, which leads to companies not adapting their production lines in time for new market opportunities, which takes time and is costly.

One of the biggest challenges of humankind has always been to maximize productive work in an efficient and effective way. To do so, it is of huge importance to plan a well-structured schedule with a detailed description of the tasks to execute, where they should be executed and when a task should be performed. This applies to areas such as transportation services, staff distribution, and, unquestionably, production systems.

Manufacturing scheduling has been studied for several decades and has been applied in industry in many different forms in an attempt to optimize the production processes and allocate resources as efficiently as possible. However, most of the scheduling has been carried out manually or using simple and limited programs that can barely improve the performance of the system. Until a few decades ago, mass production lines, producing

huge lots of the same products always following the same method, were dominant; today that is not the case, and a better planning approach needs to be considered.

In recent years, new production paradigms have been proposed to support companies tackling this problem. These paradigms, such as lean production, agile manufacturing, or smart manufacturing, have been pushed and pulled by industry and academia, which contributes to huge advances in manufacturing. However, the gap between academia and industry is still huge and the link between both of them needs to be reinforced in order to achieve consistent and acceptable results. For instance, both parts should have a closer interaction which will provide more prosperous advances, since academia is often not aware of some manufacturing requirements, such as relevant production technical features, business environment, clients preferences, or societal requirements. On the other hand, companies that do not have a solid research department may not be aware of new technologies and processes being developed, which need to be strongly matured, mainly by the contribution of industrial partners.

Now, more than ever, there is an opportunity to implement robust and efficient schedule solutions, not only on the shopfloor but even along the value chain, since there is more information available than ever before. However, there is a big challenge to implement scheduling solutions in real manufacturing systems. Despite the required complexity of such implementation in the real world, there is not a reference guide in the context of smart manufacturing to assist in the implementation of these solutions.

Consequently, the authors see the importance of establishing a baseline that can serve as a starting point when developing manufacturing scheduling systems, with the objective of developing reliable solutions that can be applied to real manufacturing scenarios.

Thus, this work aims to identify, analyze, and point out the main trends regarding the adoption of industrial requirements or constraints which academia should focus more on in order to help developing scheduling solutions in smart manufacturing systems. These requirements were identified by analyzing some relevant studies in this area to understand which requirements are being considered when implementing manufacturing scheduling solutions. Furthermore, not only the requirements were analyzed but also the objective functions, i.e., the objective the solution is trying to optimize, which can be related to production efficiency, delivery time, energetic efficiency, and so on. Consequently, further research studies may focus on how to develop scheduling solutions based on some fundamental requirements identified in this study.

The rest of this document is composed of a brief overview of manufacturing scheduling and a description of the methodology adopted in this survey. Then, the main findings are introduced, highlighting the most common scheduling requirements and which research studies are considering them. After this, a discussion about the future of scheduling in smart manufacturing environment is presented. The document ends with a brief conclusion of the article.

## 2. Manufacturing Scheduling

The recent development and advances in technology as well as the market demand for highly customized and personalized products have been pushing manufacturing companies to develop new solutions to become more dynamic and flexible to face these emergent trends and the quickly changing markets.

Most of the existing production systems are based on automated systems built to achieve high performances and high delivery rates, coming from the second and third industrial revolutions, but have no capability regarding autonomy, adaptation, and flexibility. Consequently, a group of expert technicians is needed to solve a problem each time a disturbance occurs in the production line. In addition to these restrictions, the emergence of new manufacturing paradigms, the appearance of new technologies and processes, the cheaper development of IT infrastructures, and the emerging possibility of digitization, among other factors, led to a disruption in the industrial scene.

The fourth industrial revolution—under the abbreviation of I4.0 in Europe, industrial internet or smart manufacturing in the USA, smart factory in South Korea, and made in China 2025 in China [1–3]—is happening now but started with first steps several years ago. It makes use of different emerging technologies and paradigms such as AI, cyber-physical systems, Internet of Things (IoT), cloud computing, digital twin, agent-based systems, among others, and allows the development of more dynamic and agile approaches to improve the efficiency of manufacturing systems [4,5]. The capability to collect, store, and analyze data was hugely improved, which makes it possible for manufacturers to better understand their equipment, their products, their manufacturing processes, their customers, their workers, and even their competitors, which increases the smartness degree in manufacturing systems, and leads to a better interconnection between the different systems.

Smart manufacturing comprises both horizontal and vertical integrations. Horizontal integration connects the different players of the value chain along the entire product lifecycle, which allows the optimization of the production process from suppliers to manufacturers and end users. The digitization of data and processes allow the system to be shared, analyzed, and to dynamically adapt in real-time during the manufacturing process [6]. Vertical integration combines different hierarchical levels in the production process within the factory, from working stations and human workers on the shopfloor to software technology, such as manufacturing execution systems and marketing activities [1].

However, with all the recent developments in technology, one of the main challenges in production systems continues to be the development of scheduling solutions to deal with all the planning information as well as unpredictable events on distributed production processes [7]. These challenges occur not only in the development phase, due to the implementation complexity of the systems, but also in the design phase. Nevertheless, several research articles have discussed and proposed manufacturing scheduling approaches under the smart manufacturing umbrella, such as [5,8–14].

Scheduling has been largely applied in many different areas such as energy consumption [15,16], transportation [17], staff distribution [18], and manufacturing [19,20], among other areas, to help the industries to plan their activities. For each scheduling solution for different areas of application, specific algorithms and mathematical models should be developed, since it is not achievable to develop a one-fit-all solution [18].

In manufacturing, scheduling can be considered as a process of arranging, controlling, and optimizing work on the shopfloor [21]. Sometimes, parts need to wait too long on the shopfloor due to limited resources to manage them or due to weak planning of the system. Production scheduling aims to efficiently allocate the available resources and reach a predefined goal since scheduling is a process of optimizing work and time. A scheduling problem may be described as an environment composed of one or more machines, with specific characteristics, and a set of jobs (products with one or more operations that will be processed by the machines). The goal is to optimize an objective or group of objectives by assigning each job to a specific machine in a specific time in order to be processed, while conflicts between operations are avoided [22]. Succinctly, scheduling determines what is going to be carried out and where this will happen and with what resources.

Manufacturing processes can be very dynamic. Even in environments where the processes happening on the shopfloor are always the same and known in advance, they can be affected by one or another disturbance that forces all production to stop until the problem is solved. Although some years ago manufacturing systems were not ready for this change and were not efficient enough to deal with these disturbances, nowadays, manufacturing is becoming more adaptive, dynamic, and highly flexible to meet market requirements and adjusts to every change that may improve the process. This is even more important in the era of the emergence of mass individualization, where the disturbances in the production line can be even more significant. In order to minimize the unexpected events and improve the overall production performance, one of the key challenges is to develop reliable and robust scheduling solutions. I4.0 scheduling approaches should be designed to deal with these smart and dynamic manufacturing systems and their new technologies.

Although this has been studied for decades, complex and robust scheduling solutions are frequently disregarded in real manufacturing scenarios, where they are sometimes carried out manually, on data sheets, or on simple or limited software programs. These solutions frequently lead to significant errors since they do not consider the current status of the shopfloor and are not adaptive to different scenarios. Though, more robust solutions are not implemented mainly due to the complexity of implementing them in large-scale systems with real-time constraints, since it is considered to be a non-deterministic polynomial-time hard combinatorial optimization problem which is quite difficult to reach an optimal solution for with traditional optimization techniques [23]. However, scheduling optimization has direct impacts on the production efficiency, sustainability, and also on costs of manufacturing systems and must be developed to its full capabilities [5,24].

Most researchers assume some constraints, such as that resources are always available or that the processing time of a job is known in advance and remains constant during the entire process, but in real systems this is not always true.

Disturbances may occur during the production process, which lead to a rescheduling that should be performed as fast as possible. These disturbances can be the arrival of new orders, canceled orders, or machine breakdowns which lead to the machine's unavailability, or some emergency event [5,25]. Additionally, job processing times may increase over time, which is a situation knowing as *deterioration of resources* in scheduling problems [26], or even decrease when there is a learning factor or the workload can be reinforced [27]. Consequently, to adapt to the manufacturing system, it is vital that the scheduling process is dynamic and quick to avoid unnecessary system downtimes and costs.

In flexible and agile manufacturing environments, products can have several different feasible processing plans and most of the time it is very hard to find a good one for all the products. Production scheduling is a very important decision making in a factory and it can be a difficult problem depending on the number of calculations necessary to obtain a schedule that gives an optimal or near-optimal solution for the given objectives [19].

The production scheduling optimization problem may be decomposed into several categories, according to the factory type. There are several environments depending on the machine's layout and the flow of the products, which can mainly be divided into: flow shop, which is composed of a set of machines arranged in series, one after another, where the products follow the same execution order through all the machines [28]; job shop, which can be described as a set of machines that should process a set of different jobs, where each job is composed of a group of operations to be processed in a given order, so each product may have a different route [29]; open shop, composed of machines that can perform all operations and thus there are no fixed routes for each job, which consist of unordered operations that do not have precedence constraints [30]. Usually, the essence of these is that several jobs (products with one or more operations to be processed) are assigned to a set of machines at specific times, satisfying some constraints, while trying to minimize the makespan, i.e., the time between the moment that the first job started until the moment that the last job is finished, or optimize some other objective, such as the production due dates, or the number of finished products, or the load balancing, which refers to assigning the task among different resources equally to provide better quality service, in the case of human workers, and reduce idle times and work-in-process in the case of machines [23,26,31–33].

Moreover, to produce an optimized solution, restrictions regarding product parts, material availability, machines or work capacity, start and due dates, costs, distribution requirements, or setup efficiency conditions must be known [34].

## 3. Survey Approach

To identify and characterize tendencies on the application of scheduling concepts and approaches to the manufacturing area, this study was conducted following a systematic literature review (SLR) method to reach a systematic process that synthesizes research results [35]. The procedure is synthesized in Figure 1.

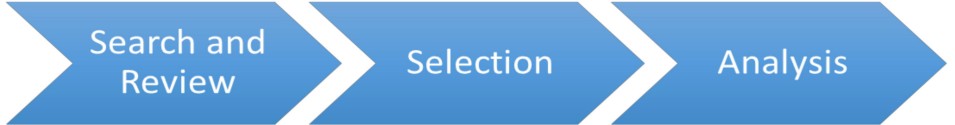

**Figure 1.** Procedure of the review.

An extended literature review was performed and the following research questions were formulated.

### 3.1. Research Questions

The research questions addressed by this work are:

RQ1 What are the constraints and targets of the manufacturing scheduling solutions found in the studied articles?

RQ2 How can scheduling systems be designed and developed so that they address different targets and requirements in the context of smart manufacturing?

### 3.2. Search Sources and Process

The first approach for finding publications related to the topic was a search on Google Scholar where it was possible to find a large number of articles related to manufacturing scheduling.

Then, for the search and selection of appropriate articles, three relevant indexed databases in manufacturing were used: Web of Science, Scopus, and IEEE Xplore. By restricting the search to these sources, some articles were automatically excluded; however, the papers indexed in these databases are expected to have more relevance, since these databases are broadly used by manufacturing and scheduling peers. Different terms were used individually or combined with others to come out with the current literature review. The most relevant terms are composed of "scheduling", "manufacturing scheduling", "production scheduling", "scheduling systems", "job shop scheduling", "dynamic scheduling", "industry 4.0", "industrie 4.0", "smart manufacturing", and "cyber-physical systems".

### 3.3. Inclusion and Exclusion Criteria

The main findings, i.e., the scheduling solutions pointed in Table 1, of this research are mainly focused on documents published during the 2013–2020 period to ensure that most information is up to date. Since the term "scheduling" is used in different applications (such as production, transportation, or staff allocation), it was necessary to exclude some papers that were not related to manufacturing. As a result, a total of 65 papers were kept as the foundation for this study.

## 4. Main Findings

In this section are presented numerous articles found in the literature that focus on different aspects of manufacturing scheduling. Contrary to traditional approaches that mostly use centralized manufacturing systems, underneath the smart manufacturing environment, most of the components are smart, autonomous, and dynamic, leading to a more intelligent and decentralized manufacturing system [36]. Consequently, a lot of data and different information need to be available in order to model and develop robust scheduling solutions.

One of the main findings is related to the innumerous different requirements that are considered among different solutions. A preliminary analysis of the selected literature allowed us to identify some requirements that may be crucial in real scenarios but are not always considered in the literature. The list of those requirements is presented next, and from here it was possible to build Table 1 and evaluate which studies consider each of these requirements.

*4.1. Manufacturing Scheduling Requirements*

Among these data, there are a lot of constraints and requirements that need to be considered before developing a production scheduling solution. Even though these requirements may be different for each particular case, some of them can be transversal to several real manufacturing scenarios. Although it is not possible to consider all of them, next are presented some relevant constraints and requirements.

### 4.1.1. Dynamic Environments

In most real-world manufacturing environments disturbances may occur over time, known as dynamic environments, and so it is important to be prepared to deal with them. This may mean that the optimal solution for the problem may also change. These disturbances may include the arrival of new jobs that need to be executed urgently, cancellation of jobs, changing of processing times, or machine availability, since machinery may be subject to maintenance operations or incur in breakdowns. Although the simplest way to cope with this problem may be to reschedule all the remaining jobs, as done by Tran and colleagues [37], it may be impractical from a temporal point of view. So, a possible solution is to the previous search space to improve the search after a change, by incorporating or removing the jobs in the previous schedule without affecting the other tasks. However, in extreme cases, reschedule could be the better option. However, in most real-world problems, changes are quite smooth, and so can be profitable to collect previous knowledge [38].

### 4.1.2. Flexibility

A flexible manufacturing system is able to produce different products by sharing tools. More factories are adopting flexible machines, which are able to perform more than one unique task; thus, a flexible scheduling solution should be adopted to those manufacturing systems. In flexible scheduling, an operation can be executed in more than one machine (routing flexibility) or each machine can be able to perform more than one operation by sharing resources (machine flexibility) [39,40].

### 4.1.3. Processing Times Variation

Most of the time, in the literature, task processing times are considered to be static, i.e., they are known in advance and do not change along the way. However, in real manufacturing systems, due to the most diverse situations, they may vary, mostly to increase over time. This situation can happen due to resource deterioration, a fault in the setup, or because of the surrounding conditions [26,41]. However, processing times may also be reduced, for example, by assigning more workforce to a task, which means that those time variations can sometimes be controlled [27]. This means that neglecting processing times may reflect in the actual production process.

### 4.1.4. Setup Times

Setup times encompass all the operations that are performed on the machines but are not related to the production process directly. It includes, for example, adding or removing product parts, calibration, machine cleaning, tests, etc. These operations can occur both before and/or after the processing of the task, and, depending on the industry, it may occur each time a new product needs to be processed. In a way to reduce setup costs, parts that need the same machine configuration may be scheduled one after the other, known as sequence-dependent setup problems [42]. However, most of the time these processes are not considered or are considered as part of the processing times of the product [27]. However, in real manufacturing systems, setup times can be a substantial part of the production time, and so should be managed wisely to generate correct information about the process. Otherwise, it can lead to incorrect information that can directly affect the production process.

### 4.1.5. Maintenance

Although often ignored in scheduling studies, maintenance activities play a crucial role in manufacturing systems, since they are a constant in real environments, either to prevent/avoid or to correct/recover failures. Even more, in the current globalized market where manufacturers focus more in reinforce the delivery reliability, sometimes at the cost of a good production and maintenance strategy [43]. Thus, maintenance activities are an important element to be considered when developing scheduling approaches, in order to have a more robust solution and achieve a better performance of the system [44,45]. Furthermore, by arranging maintenance operations strategically, companies may be able to pursue long-term competitiveness and sustainability, by providing better resources conditions, conservation, and functional life extension [46].

### 4.1.6. Precedence Activities

Even though precedence constraints have been studied for a few decades, it is still a very explored topic inside the research community. Even though it is usually assumed that every job or task to be processed is independent of any other, this is not always the case, since some jobs may be intermediates to other jobs. Thus, in the cases where there are precedence constraints, the first task of a certain job cannot start before all the tasks of its predecessors are finalized, as in the case of the assembly of two or more parts [47]. So, these constraints need to be known in advance to optimize the scheduling process.

### 4.1.7. Pre-Emption

In some cases, it may be necessary or desired that operations in jobs can be continued after a pause. This is known as pre-emption. On the other hand, when jobs cannot be interrupted, pre-emption is not allowed. Although pre-emption is rarely considered in the literature, in several scenarios, it may be needed, such as the arrival of new jobs with more importance than the ones being processed, which requires the machine to stop the operations. It can be beneficial to continue an operation in another machine or another time; unexpected cancellations by the clients might also require the stopping of production or even breakdowns in the machines as mentioned previously [27].

### 4.1.8. Release and Due Dates

Release dates (the time from when the job is available to be processed) and due dates (the time when the job must be completed) are other types of requirements or constraints present when developing scheduling solutions, although they can also be considered as part of the objectives of the scheduling. Sometimes, however, it may not be possible to complete all the jobs in the time interval between the release and due dates, but these times should not be disregarded [27]. It is important to respect the dates since the products need to be ready for delivery at some time [42] and it could be crucial to not overcome these dates. On the other hand, it could be important to not finish the products too soon as well, since this can lead to some wear in the parts or involve storage costs. Thus, when these dates are not respected, it may be necessary to apply penalties, both for early and late finishes [48,49].

### 4.1.9. Transportation

Product parts need to be moved inside the factory from one machine to another, or to the storage zone. This process can involve transportations through conveyors, robotic arms, automated guided vehicles, and many other solutions [27]. This means that first the product will not be immediately available in the next machine and a certain time is required to transport it, and second, the number of transporters is limited. So, they must be synchronized with the scheduling process along the chain or the parts need to wait for an available one, which may require to allocate each job to a transport vehicle and sequence the transport tasks to be executed by each vehicle [50]. However, transportation times are

not often considered in the scheduling problems found in the literature, which can severely affect the scheduling performance.

### 4.1.10. Storage

Another constraint that is often ignored or considered to be infinite in the literature is the storage buffer. Products may need to be placed in storage, both during and/or at the end of the production process. Obviously, this space is not infinite, and full or poorly managed storage zones may imply additional problems in the production. Thus, storage buffers may be considered when developing scheduling approaches to have more realistic solutions and reduce unexpected problems [27].

### 4.1.11. Distributed Factories

Although scheduling systems are mostly associated with the scheduling process within the factory, the scheduling process of supply chains has evolved. This increased the complexity of the problem even more, since the products may be assigned to different factories, which may be distant from each other, and the transportation across these facilities needs to be considered [40,51].

### 4.1.12. Environmental Issues

During recent years, there has been an increase in concern about the negative environmental impact caused by the manufacturing environments. Since the population is increasing, it is quite natural that the energy consumption increases as well to respond to the demand for any type of goods. Nevertheless, to achieve sustainable development in order to reduce gas emissions or acidification, it is crucial to reduce the energy demand [24,52]. This can be done, for example, by creating energy-efficient machine tools and selecting appropriate tools when acquiring them, and assess a set of key performance indicators to support the design and selection of tools [53].

As stated by [24] in their research, "none of the IPCC (Intergovernmental Panel for Climate Change) reports identifying scheduling as either a method or an instrument to improve energy efficiency [ . . . ] scheduling is rarely considered as a suitable instrument to improve sustainability either in general or concerning energy efficiency in particular". However, scheduling can be an important tool to reduce the environmental impact and achieve sustainability, since it can inform what the best steps to improve and reduce the energetic consumption and costs during the manufacturing process are, such as machines' consumption or utilization of materials [54,55].

Based on the identified requirements, the next subsection will analyze the literature considering some important requirements and restrictions that are important to consider when designing scheduling approaches to be applied in manufacturing systems.

### 4.2. Existing Approaches

In this subsection are presented some studies found in the literature covering different characteristics of manufacturing scheduling. In Table 1, some requirements the authors consider to be important in manufacturing scenarios and that contribute to improving the development of scheduling solutions are presented, which can be found in the literature.

The following abbreviations were used:

| | |
|---|---|
| ATCT—adjustment of total completion times; | MW—material wastage; |
| CJ—completed jobs; | Obj—objective function; |
| CW—cost of workers; | P—productivity; |
| DD—due date; | Pd—precedence; |
| DE—dynamic events; | PDC—total production and distribution costs; |
| DF—distributed factories; | |
| E—earliness; | PM—parallel machine; |
| EC—energy consumption; | Pr – pre-emption; |

F—flexible shopfloor;
FS—flow shop;
I—industry-oriented;
JS—job shop;
LB—load balance;
M—makespan;
MAPE—mean absolute percentage error;
MC—manufacturing cost;
MDO—maximize delivery of orders;
MeanET—Mean earliness and tardiness;
MEP—maximize early production;
MET—sum of maximum earliness and tardiness;
MFT—mean flow time;
MOO—minimize overdue orders;
MSA—maximize system availability;
Mt – maintenance;
MtC—maintenance cost;

PT—processing time;
RD—release date;
RM—resources management;
S—storage;
Set—setup times;
SCT—sum of completion times;
SM—single machine;
ST—shop type;
Stab – stability;
T—tardiness;
TDR—tardiness delivery rate;
TET—total earliness and tardiness;
TFT—total flow time;
TT—transportation time;
TV—processing times variation;
TWM—total weighted makespan;
TWT—total weight tardiness.

**Table 1.** Aggregation of solutions based on scheduling type (column 2), requirements (columns 3–17), and objectives (last column).

| Ref. | ST | I | DE | F | DF | TT | E | T | TV | Set | Mt | Pd | Pr | RD | DD | S | Obj |
|------|-----|---|----|---|----|----|---|---|----|-----|----|----|----|----|----|---|-----|
| [56] | PM | | | | | | | | | | | | | | ✓ | | TWM |
| [57] | JS | | ✓ | ✓ | | | | | | | | | | | | | M/EC |
| [40] | JS | | | ✓ | ✓ | ✓ | | | | | | | | | | | M |
| [58] | JS | | | | | | ✓ | ✓ | | | | | | ✓ | ✓ | ✓ | M/TWT/E/T/S |
| [59] | JS | | | ✓ | | | | | | ✓ | | | | | | | M |
| [60] | JS | | ✓ | | | | | | | ✓ | | | | | | | M |
| [49] | JS | | | | | | ✓ | ✓ | | | | | | | ✓ | | MET |
| [48] | FS | | | | | | | ✓ | | | | | | | ✓ | | TWT |
| [51] | JS | | | ✓ | ✓ | ✓ | | | | | | | | | | | M |
| [44] | SM | ✓ | ✓ | | | ✓ | | | | ✓ | ✓ | | | | | | M |
| [13] | FS | | ✓ | | | | | ✓ | | | ✓ | ✓ | | | ✓ | | MC/RM/ EC/TDR |
| [45] | SM | | ✓ | | | | | ✓ | | | ✓ | | | | ✓ | | MC/T |
| [61] | FS/PM | ✓ | | ✓ | | | | | | ✓ | | | | | | | M/EC/MW |
| [62] | FS | | | | | | | | ✓ | | | | | | | | M/ATCT/PT |
| [63] | JS | | ✓ | ✓ | | | ✓ | | | ✓ | | | | | ✓ | | T/EC |
| [64] | JS | | ✓ | | | | | | | | | | ✓ | | | | M/Stab |
| [65] | JS | | | ✓ | | | | ✓ | | | | | | ✓ | ✓ | | TWT |
| [25] | PM | | | | | | | | | | ✓ | | | | ✓ | | M/SCT/T |
| [66] | SM | | | ✓ | | | | | | | ✓ | | | | | | MtC |
| [41] | PM | | | | | | ✓ | ✓ | ✓ | | ✓ | | | | ✓ | | E/T/MtC |
| [67] | JS | | ✓ | | ✓ | | | ✓ | | | | | | ✓ | ✓ | ✓ | M |
| [68] | JS | ✓ | ✓ | | | | | | | | | | | ✓ | | | MFT |
| [69] | FS | | ✓ | ✓ | | | | ✓ | ✓ | ✓ | | | | | | ✓ | P/MDO/ MOO/CW |
| [70] | JS | | ✓ | | | | ✓ | ✓ | | ✓ | | | | ✓ | ✓ | | E/T/PT |
| [9] | FS | | ✓ | ✓ | | | | ✓ | | | | | | ✓ | | | T/CJ/LB |
| [47] | JS | | ✓ | | | | | ✓ | | | | | | ✓ | ✓ | | T |
| [71] | JS | | ✓ | ✓ | | | | ✓ | | | | | | ✓ | ✓ | | M/MFT/T |
| [72] | JS | | | ✓ | | | | | | ✓ | | | | | | | M/MSA/EC |
| [73] | PM | ✓ | ✓ | ✓ | | | | | | ✓ | | | | | ✓ | | T |
| [74] | JS | | ✓ | | | | | ✓ | | | | | | ✓ | ✓ | | M/TWT/MAPE |

**Table 1.** *Cont.*

| Ref. | ST | I | DE | F | DF | TT | E | T | TV | Set | Mt | Pd | Pr | RD | DD | S | Obj |
|------|----|---|----|---|----|----|---|---|----|-----|----|----|----|----|----|---|-----|
| [75] | PM | | | | | | | | ✓ | | | | | | | | M |
| [76] | JS | | | | | | | | | | | | | | | | M |
| [77] | FS | | | | | | ✓ | ✓ | | | | | | | ✓ | | MET |
| [78] | JS | | ✓ | | | ✓ | | | | | | | | | | ✓ | M |
| [79] | FS | | ✓ | | | | | ✓ | | | | | | | ✓ | | EC/T |
| [80] | FS | ✓ | | | | | | | | | | ✓ | | | ✓ | ✓ | MEP |
| [81] | JS | | ✓ | | | | ✓ | ✓ | | | | | | | ✓ | | M/MeanET |
| [82] | JS | | ✓ | | | | | | | | | | | | | | M |
| [83] | JS | | ✓ | ✓ | | | | | | | | | | | | | M/Stab |
| [84] | JS | ✓ | ✓ | ✓ | | | | | | | | | | | | | Stab/M/TFT/LB |
| [85] | JS | | | ✓ | | | | | | | | | | | | | M |
| [86] | SM | | | | | | ✓ | ✓ | | | | | | | ✓ | | TET |
| [87] | FS | | ✓ | ✓ | | | | | | | | | | | | | M/EC |
| [88] | JS | | ✓ | ✓ | | | | | | | | | | | | | M/LB |
| [89] | JS | | | | | | ✓ | ✓ | | | | | | ✓ | ✓ | | M/TET |
| [90] | PM | | | | | | | | | | | | | | | | M |
| [91] | JS | | | ✓ | | ✓ | | | | ✓ | | | | | | | M/LB/MFT |
| [26] | SM | | | | | | | | ✓ | | | | | | | | M |
| [31] | JS | | | ✓ | | ✓ | | | | | | | | | | | M |
| [92] | JS | | ✓ | | | | | | | | | | ✓ | | | | M |
| [55] | JS | | ✓ | | | | | | | | | | | | | | M/EC |
| [93] | FS | | | ✓ | | | | | ✓ | | | | | | | | M |
| [94] | JS | ✓ | | ✓ | | | | | ✓ | | | | | | | | M/LB |
| [95] | JS | | | ✓ | | | | | | | | | | | | | M |
| [96] | JS | | | ✓ | | | | | | | | | | | | | M |
| [97] | FS | | ✓ | | | | | | ✓ | | | | | | ✓ | ✓ | M |
| [8] | FS | | | | | | ✓ | ✓ | | | | | | | ✓ | | M |
| [98] | JS | | ✓ | | | ✓ | | | | | | | | | | | EC/MC |
| [99] | JS | | ✓ | | | | ✓ | ✓ | | | | | | | ✓ | | M/MeanET |
| [100] | JS | ✓ | ✓ | | | | ✓ | ✓ | | | | | | | ✓ | | MET/PDC |
| [101] | FS | | ✓ | ✓ | | | | ✓ | ✓ | | | | | | ✓ | | T |
| [37] | JS | | ✓ | | | | | ✓ | | | | | | | ✓ | | M/T |
| [102] | FS | | | | | | | | | | | | ✓ | | | | M |
| [42] | PM | ✓ | | | | | | ✓ | | ✓ | | | | | ✓ | | T |
| [32] | PM | | | ✓ | | | | | | | | | | | | | LB |

As [23] noticed in their research survey on AI strategies for resolving job shop scheduling problems, only a small percentage (8.06%) of the studied researchers published articles focused on solving real-life industrial problems, between 1997 and 2012. Additionally, [103] stated in their research survey study (focused on articles between 1990 and 2014) that most work was focused on testing the developed algorithms on benchmark instances, and just a fraction of the research has been applied to practical problem solutions as compared to pure research.

The data presented in Table 1 were processed using Microsoft Excel, taking advantage of the calculation and graphic tools features available.

From the 65 articles analyzed during this work (Table 1), it was possible to observe that only nearly 10% tried to solve the scheduling problem in real industrial scenarios. The focus on algorithm development is of huge importance and can contribute greatly to solve real problems. However, real scenarios have an entire set of conditions and circumstances that are not considered when algorithms are developed in the laboratory.

The current problem in obtaining feasible solutions for smart manufacturing scenarios is not related to technology by itself [104], as technology has evolved a lot during recent decades, it is related to managing all the actors and the connections between them, and the use of them to improve the industry. Thus, more effort should be dedicated to solving industry-oriented problems.

In addition to this, regarding the objectives, most researchers considered, i.e., tried to solve, single-objective problems, while some of them tried to optimize at least two or three objectives, as shown in Figure 2. In the case of Figure 2 a counting was performed for each number of objectives used in each article divided by the total number of articles. For example, when analyzing which articles considered two objectives, it is only necessary to count the number of articles with two objectives (16) and divide by the total number of articles (65), which is approximately 25%. For Figures 3 and 4 a similar approach was adopted.

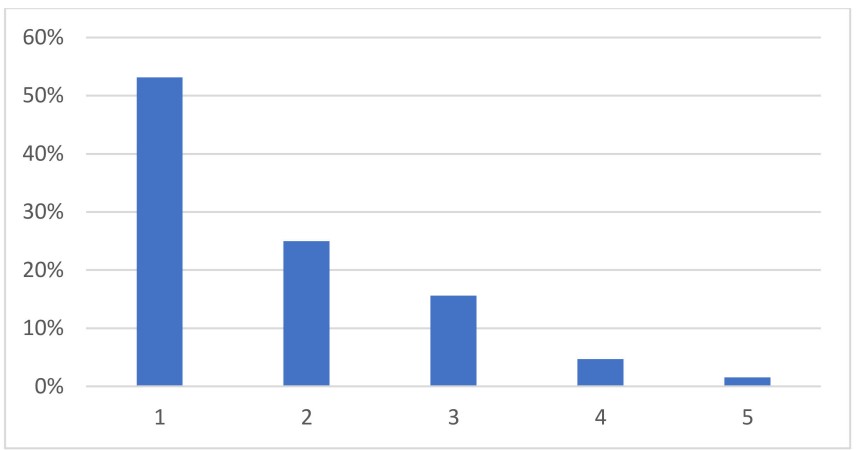

**Figure 2.** Number of objectives considered per study.

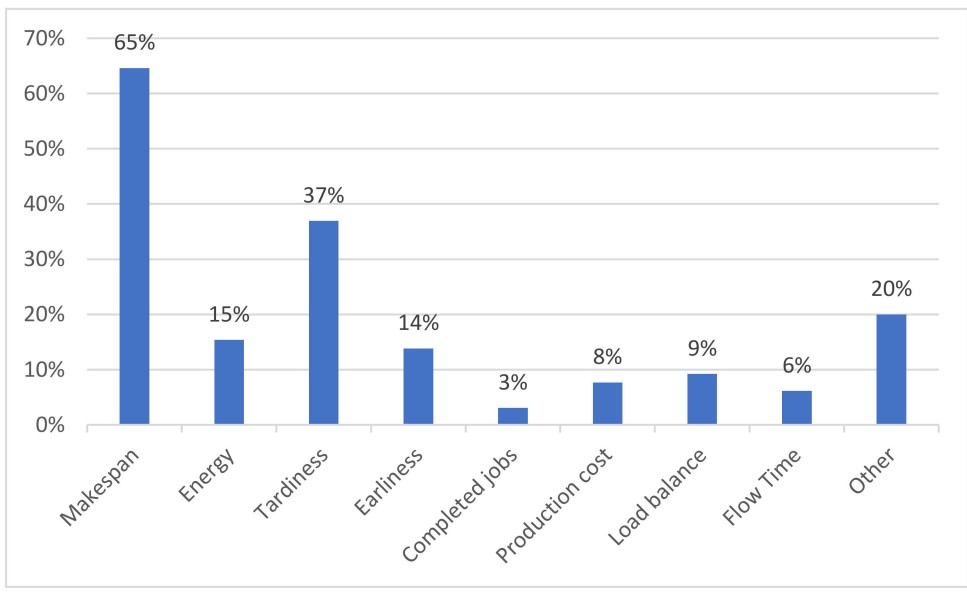

**Figure 3.** Percentage of articles considering each objective.

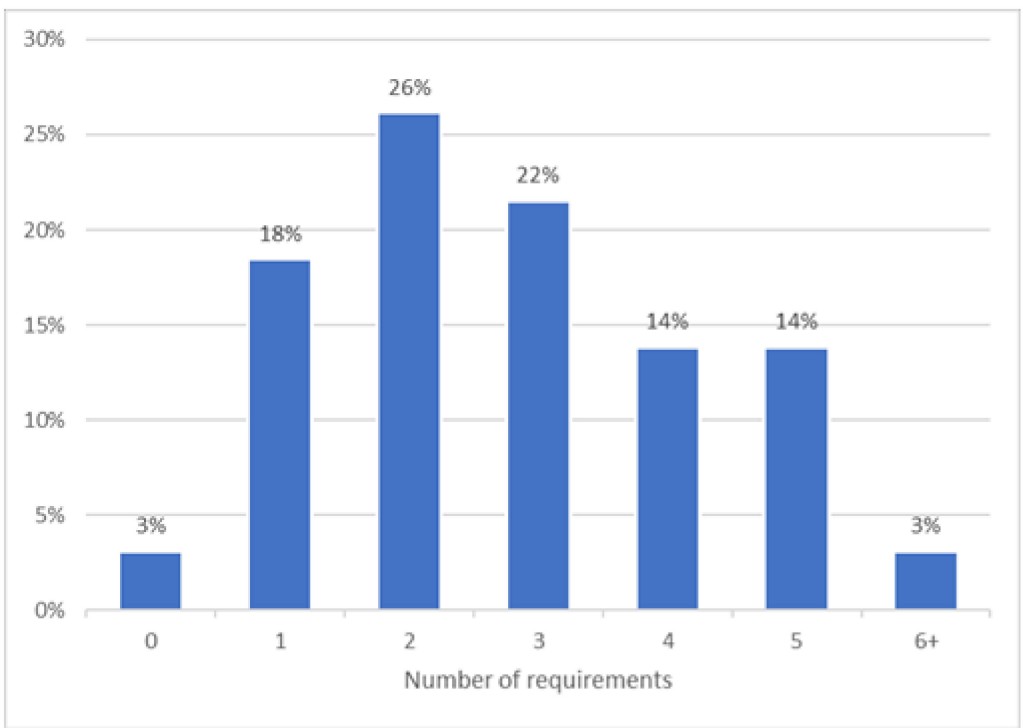

**Figure 4.** Number of requirements considered per study.

Within these objectives, most researchers, around 65%, focused on trying to optimize makespan, while more than a third try to deal with problems related to the tardiness of finished products, as it is possible to observe in Figure 3. The *Other* category is composed of features with weights of less than 2% each and so they were joined together. One outstanding point is that in the third position, 15% of the articles considered energy consumption issues, which makes clear that is a concern in the research community regarding environmental issues and sustainability. Nevertheless, although some authors use reference benchmarks for testing and comparison, it is not clear which techniques are superior to others for a specific problem, although they can be better than others in specific points. This leads to solutions that are good at beating benchmarks but are not able to be deployed in real manufacturing systems [103].

In addition to this, some problems arise when unrealistic assumptions are made since manufacturing environments are very dynamic and do not rely on static features. These assumptions can rapidly lead to unfeasible scheduling solutions which can be costly to the companies [27]. The most common assumptions are: all jobs and machines are available at starting time and release times are not considered; each machine can only execute one specific operation, which is not always true since some machines may have the flexibility to perform more than one operation, by changing tools, for example; the processing time of an operation is unchangeable, but processing times may change according to the conditions in a specific moment; machines never break and they are always available; setup times of any operation are sequence-independent and are included in the processing time, which is not always true and may compromise the entire schedule; pre-emption is not allowed; storage buffers are unlimited.

Aligned with this problem, the authors of [27] state that " . . . the intersection among three or more sets of constraints for any shop layout would return an empty set of references most of the time". This means that only a few studies contemplate more than three constraints. From Table 1, it is possible to observe that only about a third of the research studies considered more than three of the identified requirements in the same approach, as demonstrated in Figure 4.

It is also possible to observe that the most adopted requirements are flexibility, related to the ability of the shopfloor to adapt and adopt different features, and due dates, related with the date when each job shall be completed. After this comes the consideration of dynamic environments, where the arrival of new jobs, job cancellations or machine breakdowns during the execution process are considered and a reschedule may be necessary. Finally, the last consideration is tardiness, which evaluates the extent to which jobs are delayed from the initial deadline. All the other requirements are less expressive, being adopted in less than 20% of the articles.

This is a substantially small number considering all the requirements and constraints that can be found in manufacturing systems. Additionally, almost half of the studies considered two or three constraints. On the other side, two of the studied articles did not consider even one of the requirements mentioned in this study, as they only tried to optimize the objectives without any kind of limitation.

## 5. Scheduling in the Context of Smart Manufacturing and Next Steps

### 5.1. Gaps and Challenges

Sometimes it can look as researchers are facilitating when trying to narrow the bridge between academic studies and industry, as the published studies are often too simple, vague, and even have convenient restrictions and simplifications that do not reflect real industrial systems. This may be true. However, on the other hand, both computational and real-world complexity can difficult the problems solving. Additionally, most of the time companies are not willing to provide sensitive information that can be valuable for developing better manufacturing systems. These days, where data are becoming the core of industrial systems, it is crucial that data are provided to academia, so better solutions, not only for the shopfloor but for all value chains, may emerge. Nonetheless, academia should also make an effort to meet the industry's needs. Although there are plenty of scheduling studies in the literature, the research dealing with real-world problems is very uncommon. Likewise, most approaches do not allow solutions to be scalable and to be reused in some different scenarios.

Thus, the main challenges, and similarly the main gaps, in manufacturing scheduling research for smart manufacturing comes from the fact that most studies found in the literature, as mentioned before, are based on assumptions, related to the manufacturing environment, that are, commonly, not true and even naïve in real manufacturing systems. Furthermore, to cope with more realistic scenarios, more constraints and requirements need to be considered simultaneously to replicate as well as possible real manufacturing systems.

### 5.2. Human Factor

Based on the study presented, it is possible to identify that the objectives and requirements related to humans and the operators are not usually taken into account by manufacturing scheduling solutions. However, it is getting more and more usual for companies and society to explore how these new smart manufacturing solutions must deal with humans and how they can or cannot help humans within the industry. Hence, the authors believe that in the near future, it is necessary to explore new manufacturing scheduling approaches where these systems could use human-related aspects such as the available human resources or tiredness of the operators could be used to optimize the systems not only from the production or energy efficiency point of view but also taking into account these aspects to improve the integration of these systems and the operators, developing more harmonized workplaces.

The necessity to develop systems, including the human aspects, is critical for future research activities from the sustainability point of view. This aspect is one of the three pillars of sustainability. However, the addition of the human factors for manufacturing scheduling studies highly increases the studies' complexity since all humans are different and have different behaviors for the same situations. So, the authors believe that it is crucial to create interdisciplinary teams to explore this aspect because such systems' development will

require analysis and understanding of elements that usually are far from the manufacturing scheduling experts' expertise. Simultaneously, this topic constitutes a relevant topic to be explored in the future due to the necessity for these new smart manufacturing-oriented solutions to be aware of the operators and their behaviors, taking that into account during the optimization process.

### 5.3. Opportunities Future Work

Nowadays, manufacturing systems are becoming enormous living robotic environments, and, in some cases, there is limited human presence. There are smart products, which constantly give feedback about their own status to the system, and smart resources, which can do the same and inform the system in real-time if some problem occurs in the production line. This intelligence within the system allows the different components to communicate and interact in order to reach common objectives. Nevertheless, on the other side, it also allows the extraction of more data and information from the system than ever before and transforms that information into useful knowledge. Knowledge can be used to improve multiple strands of the system, not only within the factory but also across the entire supply chain. This fact can immensely improve production scheduling systems, which are able to extract dynamically and in real-time more and more data from a vast variety of smart components, which can help to provide more robust and efficient scheduling solutions. However, in recent years there has been more and more interoperability between tools on shopfloors and existing legacy solutions are not able to account for this.

Although there is a huge complexity involved in developing smart manufacturing systems and there may be some restrictions regarding the process and memory power, nowadays the main limitations are not related to the hardware nor to the connection between entities since the emergence of cyber-physical systems has allowed all kind of entities on the shopfloor to be virtualized and connected together, thus permitting the easy execution of monitoring and controlling activities in the production line. One of the main problems is related to the linking between different types of data sources and how they need to cooperate to achieve better and more efficient performances. Consequently, it is necessary to go one step further and direct more efforts towards modeling, optimization, and standardization of manufacturing systems [105].

Therefore, to harmonize the designing and development of manufacturing systems, some reference architectures have emerged over recent years, such as ISA-95 [106], 5C architecture [107], Smart Grid Architecture Model—SGAM [108], Industrial Internet Reference Architecture—IIRA [109], or Reference Architectural Model for Industrie 4.0— RAMI4.0 [110]. These architectures aim to ensure a common comprehension, achieve standardization, enable semantic interoperability, and provide consistent operation models for the system. By adopting a reference architecture in manufacturing scheduling it will be possible to manage all the information coming from different sources in a consistent and homogeneous way and apply a core scheduling solution to different scenarios.

One of the possible ways to achieve this is to create an Asset Administration Shell (AAS) for scheduling and consequently for all other entities related to scheduling, under the RAMI4.0 architecture. An AAS consists of transforming physical components such as robots, machines, or devices and, similarly, intangible assets such as functions, plans, or an entire network into Industrie 4.0 components, which will then allow standardization as much as possible for solutions in engineering, operation, and management, and implementation of a heterogeneous communication structure in a smart manufacturing-oriented system. The AAS is the virtual representation of the asset that encompasses all the information and technical functionalities of the asset and manages communications with other Industrie 4.0 Components [110]. Through describing and modeling the asset components in smart manufacturing environments aligned with RAMI4.0 architecture, it will be possible to standardize and optimize the scheduling development process. In sequence, by clarifying which components should interact in the scheduling process and which data

need to be available and flow within the system, scheduling designers and developers may be better prepared, and companies may benefit from the scheduling solutions created.

## 6. Conclusions

This work provides a literature review on smart manufacturing and, more specifically, on manufacturing scheduling. This is a very explored and discussed topic, which can hugely contribute to develop better manufacturing systems and improve the overall performance of those systems, regarding time, throughput, resources, or energetic optimization, among others. However, scheduling designing and development is not structured and harmonized between different entities, since it usually developed specifically for each case.

Consequently, the objective of this study is twofold. On one hand, the aim was to investigate which requirements and constraints are fundamental considerations when developing scheduling solutions for industrial scenarios. This was explored in subchapter 4.1 by analyzing some reference articles and books on the topic where requirements were used more commonly and their role in the scheduling process. On the other hand, this study analyzed several articles to identify the previous requirements they considered and which objectives they tried to optimize, which can render them robust approaches to be deployed in real manufacturing situations. The result of the last point came up as table containing all the 65 analyzed articles, pointing out which requirements they considered and the most common optimization objectives.

The results show that the vast majority of the articles only consider one or two objectives, mainly the makespan, which was found in around 65% of the articles, and tardiness related objectives, which were found in 37% of cases. It is also important to note that around 15% of the articles focused on solving energy consumption related issues, which shall be a major focus during these times of environmental awareness. Actually, environmental issues have been playing an increasingly important role in manufacturing scheduling, where studies are mainly making energy consumption assessment. However, in real manufacturing systems, depending on the case, it may be necessary to consider multiple optimization objectives. Furthermore, regarding the number of requirements considered in each study, it is possible to observe that most of the articles considered two or three requirements per case, while around 17% considered five or more constraints. It is important to understand that the more requirements considered, the more robust a scheduling solution can become, and then the more applicable it will be in real scenarios. As far as the authors know, this kind of study is not to be found in the literature.

In the analyzed literature, no study was found describing the components of manufacturing scheduling for reference architectures. Different approaches consider different constraints, and there is not a common and uniform way of developing scheduling solutions, even though there are common points in industrial systems that may be harmonized. Having a reference for scheduling designing and development can speed up the creation of scheduling solutions and make it easier to adapt these solutions to different scenarios. Knowing that scheduling can have a direct impact on production efficiency, sustainability, and costs of manufacturing systems, it is of huge interest to conduct in-depth research on how to model smart components to optimize scheduling approaches in smart manufacturing systems. Providing a structured model that includes relevant information about the inputs of the scheduling process and the desired outputs has great relevance for the design of scheduling solutions to cope with the smart manufacturing paradigm.

**Funding:** This work was supported in part by the FCT/MCTES project CESME—Collaborative & Evolvable Smart Manufacturing Ecosystem, funding PTDC/EEI-AUT/32410/2017.

**Institutional Review Board Statement:** Not applicable.

**Informed Consent Statement:** Not applicable.

**Data Availability Statement:** Data sharing not applicable.

**Conflicts of Interest:** The authors declare no conflict of interest.

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
