# Peer review of "Smart Manufacturing Scheduling Approaches—Systematic Review and Future Directions"

_applsci, doi:10.3390/app11052186_

Round 1
Reviewer 1 Report
This is set as a review article. However, the analysis in Table 1 and in Figures 2-4 add value as an independent research. The coverage of contemporary references is solid. 69 references form overall 101 are from last five years. Important research trends were analysed in manufacturing scheduling.
Author Response
The article is indeed a review article. The analysis in Table 1 and in Figures 2-4 reflect what was analyzed during this work and present a summary of the major outcomes of this research, where the main objective was to understand which kind of requirements and constraints can be found in manufacturing environments and which ones are being applied in academic studies
Reviewer 2 Report
Dear authors,
In the attachment you can find the review report of the paper.
Best regards

Reviewer 3 Report
Dear authors, judging by the materials presented, your article is undoubtedly relevant and reflects the modern understanding of the latest advances in technology and the demand for products with a high degree of individualization.
Dear author, the fact that your article is unnecessarily overloaded with theoretical material presented in text form raises a question, and this complicates the perception and analysis of your contribution.
I think that there is a need to bring the revision and structuring of literary sources on this issue. Regarding the sampling of experimental data, it is necessary to process all the information using modern statistical software tools, and show for the reader of your article that your research is sufficiently reliable.
It is necessary to visualize some of the information for a better perception of a potential reader.
It is necessary to provide research data on the use of your developments in real conditions.
